# Botanical Composition and Species Diversity of Arid and Desert Rangelands in Tataouine, Tunisia

**Mouldi Gamoun**  **and Mounir Louhaichi ***

International Center for Agricultural Research in the Dry Areas (ICARDA), 2049 Ariana, Tunisia;
M.Gamoun@cgiar.org
* Correspondence: M.Louhaichi@cgiar.org; Tel.: +216-7175-2099

**Abstract:** Natural rangelands occupy about 5.5 million hectares of Tunisia's landmass, and 38% of this area is in Tataouine governorate. Although efforts towards natural restoration are increasing rapidly as a result of restoration projects, the area of degraded rangelands has continued to expand and the severity of desertification has continued to intensify. Any damage caused by disturbances, such as grazing and recurrent drought, may be masked by a return of favorable rainfall conditions. In this work, conducted during March 2018, we surveyed the botanical composition and species diversity of natural rangelands in Tataouine in southern Tunisia. The flora comprised about 279 species belonging to 58 families, with 54% annuals and 46% perennials. The Asteraceae family had the greatest richness of species, followed by Poaceae, Fabaceae, Amaranthaceae, Brassicaceae, Boraginaceae, Caryophyllaceae, Lamiaceae, Apiaceae, and Cistaceae. Therophytes made the highest contribution, followed by chamaephytes and hemicryptophytes. Of all these species, 40% were palatable to highly palatable and more than 13% are used in both traditional and modern medicine.

**Keywords:** vegetation; species richness; drylands; south of Tunisia



## 1. Introduction

Climate change and human activity represent a big threat to biodiversity [1–3]. The continuous damage to biodiversity increases the rate of species extinctions, which undermines our capacity to combat desertification, reduce poverty, increase food security, and exclude invasive species [4–7]. The arid rangelands are among the most important ecosystems and provide a great variety of services and homes to pastoral and agro-pastoral communities. In particular, they cover diverse habitats and ecological communities. They are also economically important given the tremendous richness of edible plant, forage, medicinal, and economic species. The effects of any disturbance on plant biodiversity of arid rangelands are more apparent and more profound than in other ecosystems. Several studies have shown the negative effects of grazing, drought, and other human activities on biodiversity and plant species of arid rangelands [8–11]. In the face of the degradation of natural resources and the progress of desertification, maintenance of biodiversity through active management has recently become an important challenge for biodiversity conservation [12]. Restoration efforts for natural rangelands are important and result in high species diversity [13–17]. Some efforts have been implemented to restore rangelands functions, and resting is the most widely and successfully used practice in arid rangelands restoration and represents an ideal still applied in rangelands management. Floristic richness is considered to be particularly valuable and a prime target for establishing conservation priorities [18–20]. An essential part of many studies on rangelands restoration and sustainable management is the medicinal use of plants and their economic value, which requires knowledge of the botanical composition and species richness present in natural rangelands.

In 2017, the botanical composition of the arid and desert areas of southern Tunisia was assessed, during which the area received an average rainfall (100 mm) and recorded 135 species [16]. More recently, the arid and desert areas of southern Tunisia witnessed an

exceptional wet season, causing a magnificent phenomenon called the "flowering of desert rangelands", which led to the blossoming of a wide range of flowers during the month of March 2018. Under these conditions, arid and desert rangelands show a great resilience illustrated through a very high plant diversity.

These rare favorable conditions offer a golden opportunity to record the greatest plant diversity in arid and desert rangelands and identify key species that can survive in this ecosystem. Specifically, the objectives of our study are to examine the botanical composition, including plant family, life form, habitat class, palatability, and medicinal and aromatic plants, and to determine the relevance of plant diversity to arid and desert rangelands stability and functioning. It is important to understand the significance of these rangelands in terms of providing several ecosystem services of vital importance for the local communities. Our findings could contribute towards developing holistic and sustainable rangelands management practices and finding innovative solutions to protect key natural habitats in southern Tunisia and similar arid environments.

## 2. Materials and Methods

### 2.1. Study Area

Tataouine is located in the extreme south of Tunisia and it has a unique geographical location bordering Libya and Algeria (32°55′32″ N, 10°26′39″ E). It is the largest governorate of the country, covering an area of 38,889 km². Tataouine shares borders with the governorates of Medenine and Kebili to the north, Libya and the governorate of Medenine to the east, and Algeria to the west (Figure 1).

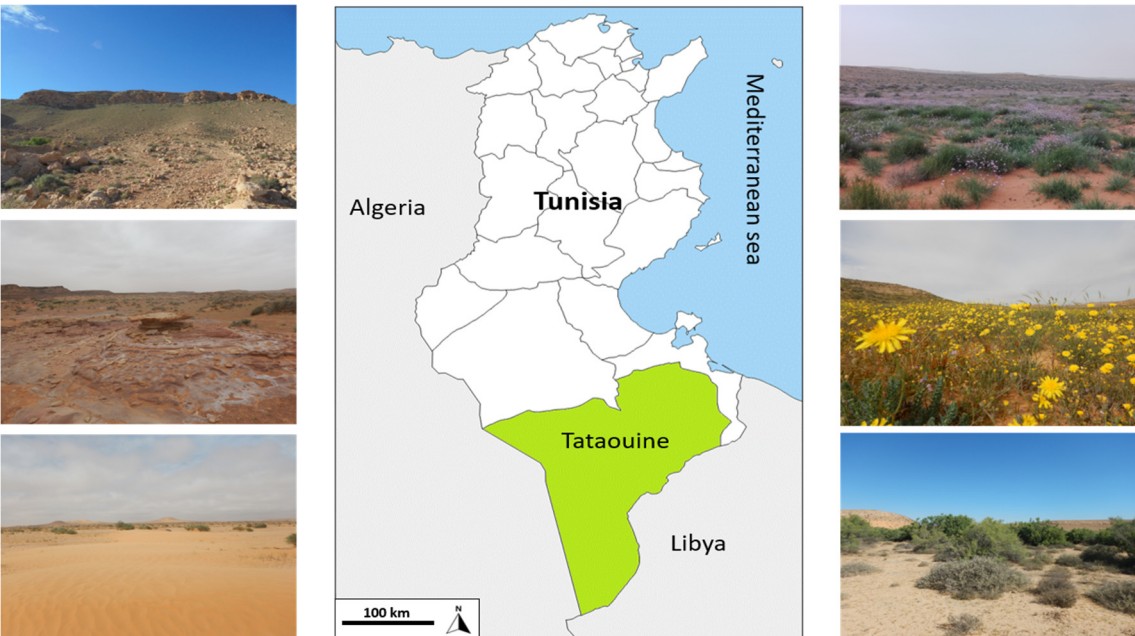

**Figure 1.** Map of the governorate of Tataouine and its location in Tunisia; pictures taken during spring 2018 showing the different natural habitats exploding with flowers in a rare phenomenon.

The total annual average rainfall in the area is below 100 mm, with cold, dry, and windy winters and hot and arid summers. Sand dunes and sandy, limestone, and gypsum soils are the most common soils in the desert area. A total of 38% of Tataouine consists of natural rangelands (1,500,000 ha), representing more than 27% of the total rangelands in Tunisia, grazed by an estimated 390,000 head of sheep, goats, and camels [21]. The vegetation is scattered and dominated by shrubs adapted to drought and extreme temperatures.

### 2.2. Data Collection

As a consequence of the favorable rainfall conditions recorded in southern Tunisia during 2017/2018 (Figure 2), vast rangeland areas were covered by hundreds of bloomed plant species. The present inventory represents the highest species richness recorded in the region of Tataouine. The inventory was conducted through frequent field visits made to various rangeland sites from March 2018 to April 2018 to identify different plant species. Since the areas is huge, we had to rely on expert knowledge, including herders who practice transhumance, as well as elderly pastoralists, to guide us to the exact locations where certain species were resurfacing during such an exceptional favorable year. All species were photographed using a high-resolution digital camera, showing structure, leaf, stem, flower, and fruit, if existing. All recorded plants were then identified with the help of available Tunisian flora of Pottier-Alapetite [22,23]. The new nomenclature for inventoried species was updated using the synonymic index of the flora of North Africa by [24] and POWO (Plants of the World Online) [25]. Once plants were identified, family, life form, habitat class, and livestock palatability index (PI) of each species were determined.

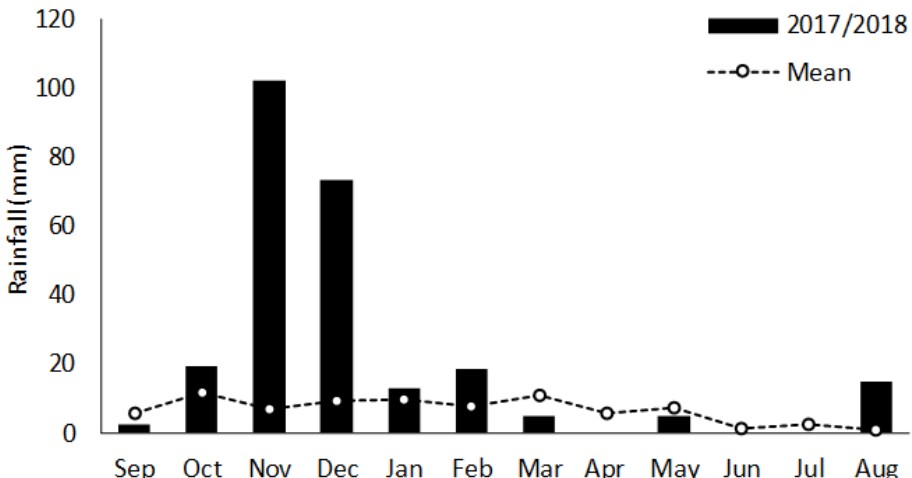

**Figure 2.** Annual and monthly rainfall variability during the season of 2017/2018 in Tataouine, southern Tunisia [26].

### 3. Results and Discussion

Several studies have claimed that precipitation has a significant impact on the vegetation dynamics of arid rangelands [16,27–29]. More particularly, rainfall variability has a great impact on plant phenology [30–35], plant life cycles [36–40], and therefore species richness [41–44].

Rainfall distribution and quantity play important roles in encountering the maximum number of plant species that can grow in arid and desert rangelands. The favourable conditions of 2017/18 resulted in a large number of plant species across the rangelands of Tataouine, many of which were not recorded in recent decades, such as *Helianthemum crassifolium*, *Helianthemum ruficomum*, *Helianthemum virgatum* subsp. *africanum*, *Plantago afra*, *Dactylis glomerata*, *Andrachne telephioides*, *Catananche arenaria*, *Coris monspeliensis*, and *Teucrium alopecurus*. A total of 279 species were recorded, belonging to 58 families (Supplementary Materials). The altered climate conditions in the south of Tunisia were associated with increased species richness. The families with the highest number of species were Asteraceae with 43 species (15.35%), Poaceae with 30 (11%), Fabaceae with 24 (8.57%), Amaranthaceae with 16 (5.71%), Brassicaceae with 16 (5.71%), and Boraginaceae, Caryophyllaceae, and Lamiaceae with 12 species each (4.28% each). These families were the most diverse of the flora in the arid rangelands of Tunisia (Figure 3).

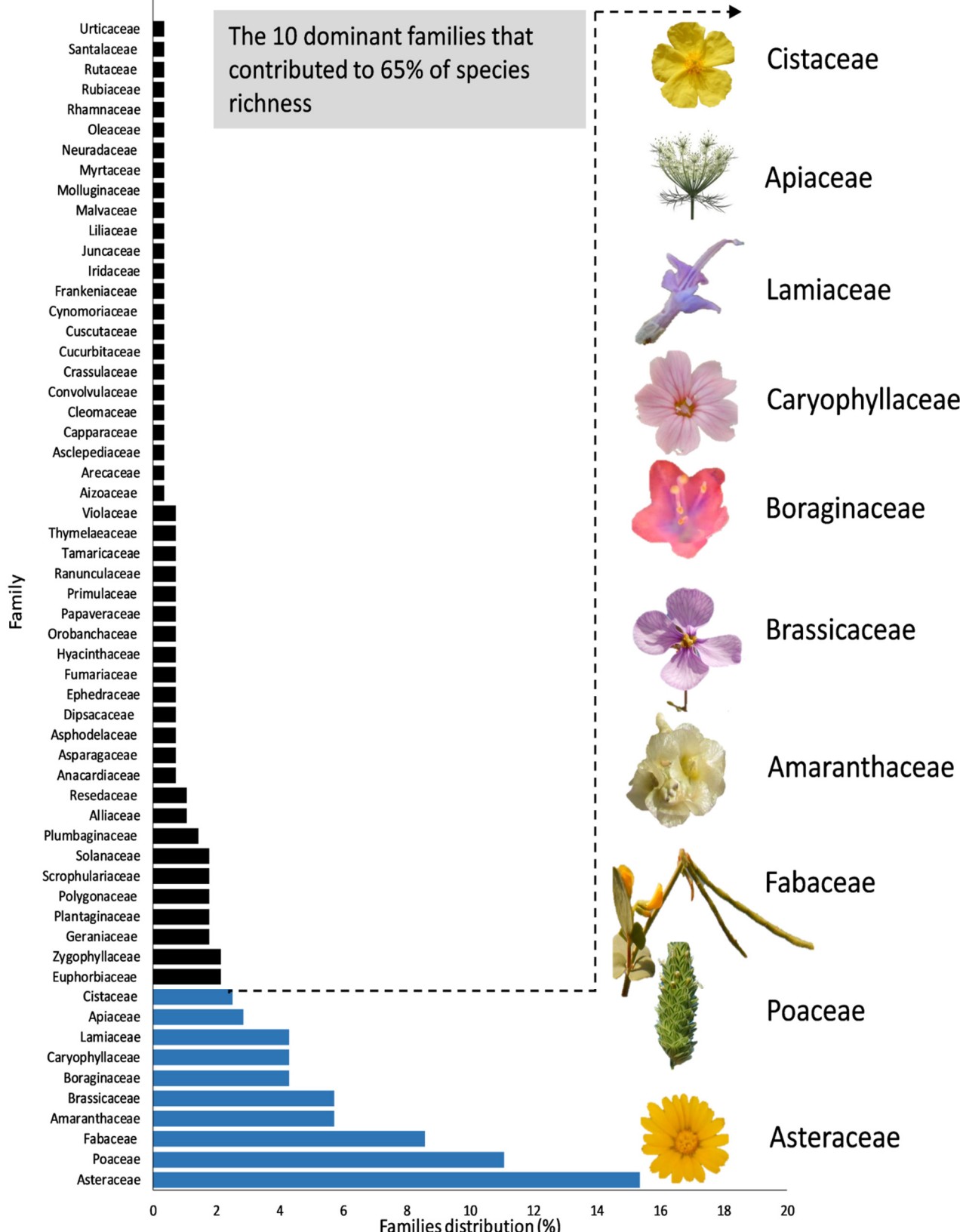

**Figure 3.** Plant family diversity in the arid and desert rangelands of Tataouine, Tunisia (spring 2018).

Overall, 54% of the species were annual (150 species) and 46% were perennial (130). Several annual plant species can be found exclusively in improved microsite conditions (e.g., lower temperatures, reduced solar radiation, or increased organic matter) in rocky mountains and benefit from higher rainfall. For example, the presence of *Lamarckia aurea* and *Umbilicus rupestris* indicates favorable environmental conditions, which may be the result of available microsites for plant establishment under higher rainfall conditions.

During the extremely wet year, the arid and desert rangelands, with all their different habitats (mountain, plain, wadi, dune), tended to be dominated by therophytes (ephemerals and annuals) both in grazed and managed areas [45]. These species germinate on conditions that are favorable and thrive under heavy rainfall. The life-form spectrum of these recorded species showed that 48% were therophytes, 20% were chamaephytes, 20% were hemicryptophytes, 5% were nanophanerophytes, 3% were geophytes, 2% were macrophanerophytes, 1% were phanerophytes, and 1% were parasites (Figure 4). The dominance of therophytes can be attributed to the large number of microsites suitable to annual plants that have rapid germination and growth, thus increasing their abundance [46–48]. Chamaephytes can survive in arid rangelands because they are highly adapted to arid conditions [49–54]. The perennial shrubs were mainly chamaephytes, such as *Haloxylon schmittianum*, *Haloxylon scoparium*, *Helianthemum kahiricum*, *Helianthemum lippii*, *Rhanterium suaveolens*, and *Gymnocarpos decander*, which characterize the dry and desert rangelands. Hemicryptophytes were very common plants in large areas of arid rangelands, with more than 50 species, most of which were Poaceae that emerge from seeds and propagate vegetatively from plant parts [55].

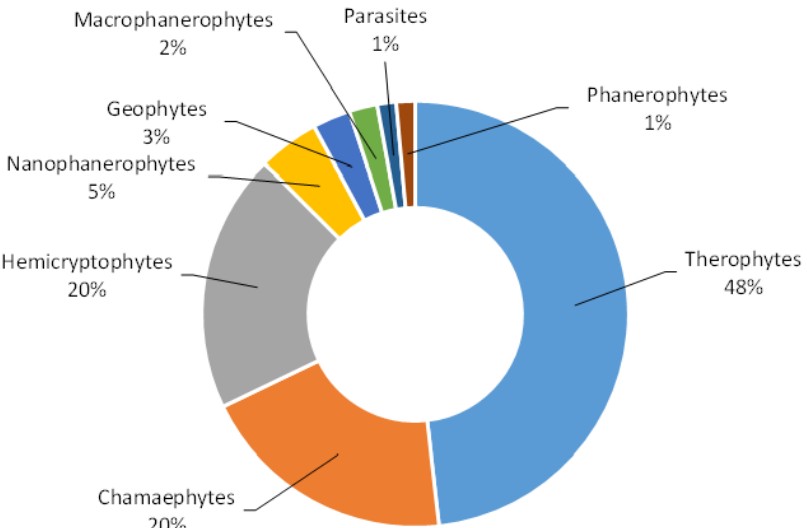

**Figure 4.** Life form distribution of plant species in the arid and desert rangelands of Tataouine, Tunisia (spring 2018).

During the rainy season, the growth of herbaceous plants in the rangelands of Tataouine was very important (Figure 5). Herbaceous vegetation was dominant, with 73% of species, followed by shrubs 25%, and trees 2%, with climbers represented by only one species: *Convolvulus supinus*. Not surprisingly, herbaceous species were very abundant because the majority of herbaceous species are therophytes and hemicryptophytes, which dominate the rangelands.

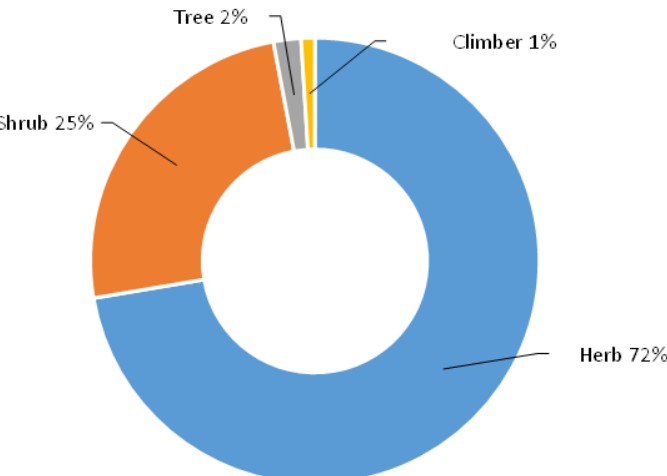

**Figure 5.** Habitat class of plant species in the arid and desert rangelands of Tataouine, Tunisia (spring 2018).

These shrubs growing in arid rangelands are a vital source of food for livestock and wildlife; however, species differ widely in their palatability [16]. In North Africa, [56–59] proposed a PI scale for forage species with six classes: highly palatable (5), very palatable (4), palatable (3), fairly palatable (2), occasionally palatable or poorly palatable (1), and not palatable (0). The inventoried rangelands had a relatively high species richness, due to its diverse plant communities combined with favorable rainfall conditions. Rainfall masked the effect of livestock grazing and drought on botanical composition during the spring of 2018. There were more palatable than unpalatable species (Figure 6).

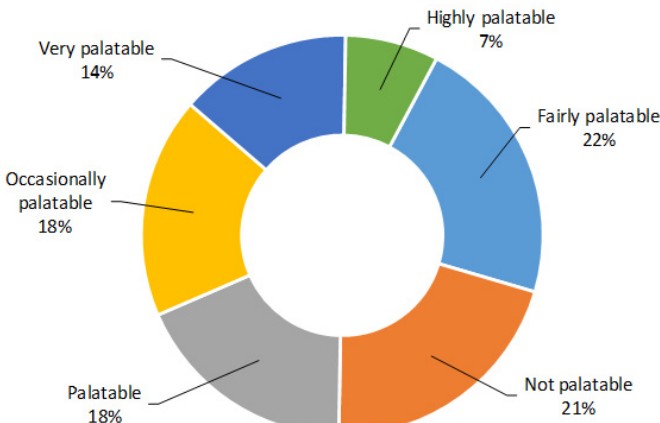

**Figure 6.** Palatability of plant species in the arid and desert rangelands of Tataouine, Tunisia (spring 2018).

More than 110 palatable species (40%) in the local flora of natural rangelands of Tataouine have been sources of foraging for sheep, goats, and camels since ancient times. Among the palatable species, 7% (21 species, Figure 7) were highly palatable and 14% (39 species) were very palatable, while 22% (61 species) were fairly palatable and 18% (50 species) were occasionally palatable or poorly palatable. However, 21% of the flora (58 species) were unpalatable or toxic species. The most valuable native perennial shrub species that were highly palatable included *Anabasis oropediorum*, *Argyrolobium uniflorum*, *Echiochilon fruticosum*, *Gymnocarpos decander*, *Helianthemum ruficomum*, *Helianthemum virgatum*, *Helianthemum lippii* var. *intricatum*, and *Helianthemum lippii* var. *sessiliflorum*. In addition to its high palatability, the shrub *Helianthemum lippii* var. *sessiliflorum* is the main host of truffles (*Terfezia boudieri*, *Tirmania pinoyi*, and *Tirmania nivea*, known locally as Terfes), which has economic importance and is regarded by the local population as a great delicacy;

it is collected from rangelands and sold in local and foreign markets. Additionally, the presence of this important component of the mycological flora in arid and desert rangelands is a good indicator of rangelands health. Truffles develop underground after heavy autumn rain and parasitize the mycorrhizae associated with the roots of *Helianthemum* spp. [60].

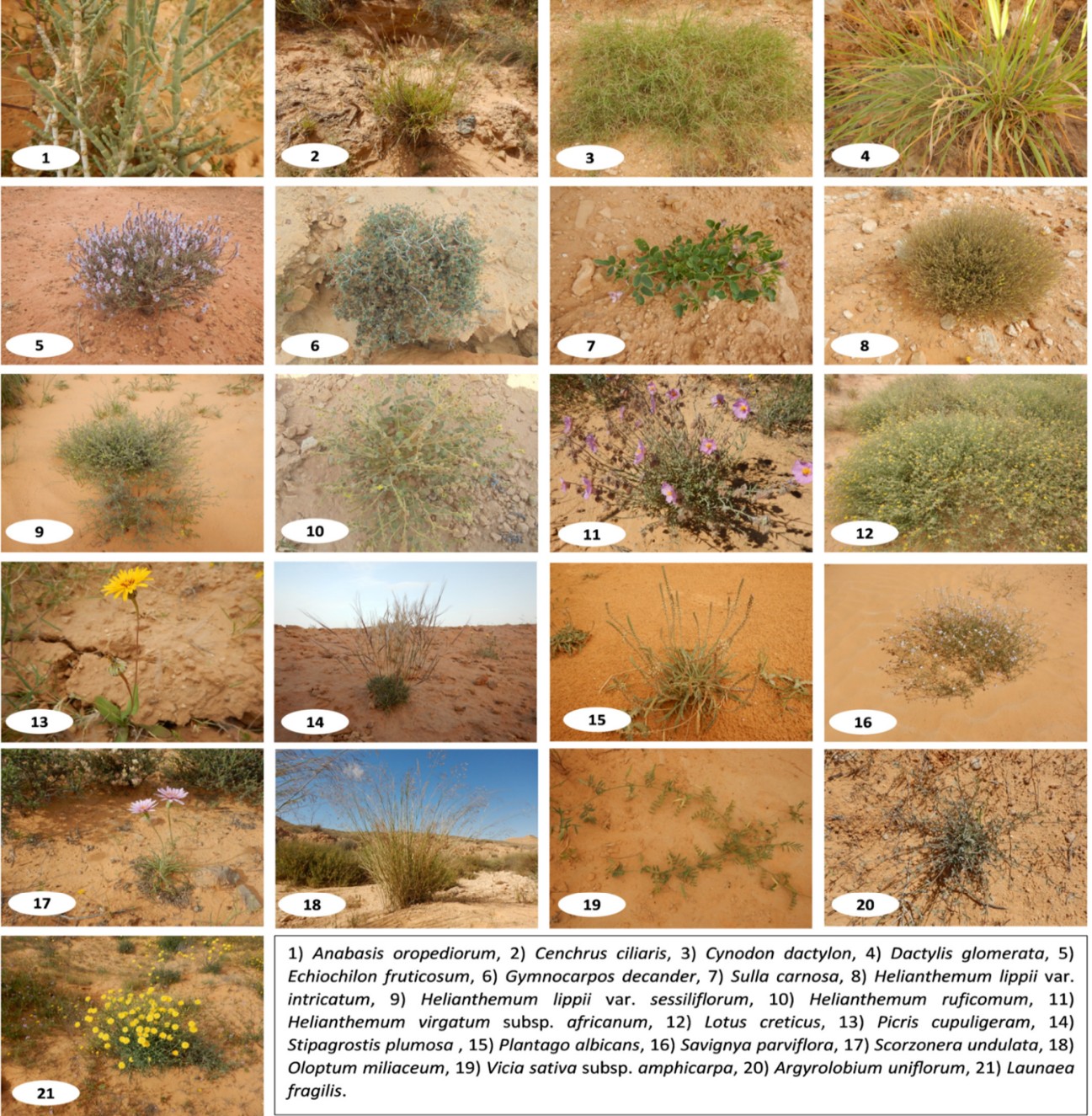

1) *Anabasis oropediorum*, 2) *Cenchrus ciliaris*, 3) *Cynodon dactylon*, 4) *Dactylis glomerata*, 5) *Echiochilon fruticosum*, 6) *Gymnocarpos decander*, 7) *Sulla carnosa*, 8) *Helianthemum lippii* var. *intricatum*, 9) *Helianthemum lippii* var. *sessiliflorum*, 10) *Helianthemum ruficomum*, 11) *Helianthemum virgatum* subsp. *africanum*, 12) *Lotus creticus*, 13) *Picris cupuligeram*, 14) *Stipagrostis plumosa* , 15) *Plantago albicans*, 16) *Savignya parviflora*, 17) *Scorzonera undulata*, 18) *Oloptum miliaceum*, 19) *Vicia sativa* subsp. *amphicarpa*, 20) *Argyrolobium uniflorum*, 21) *Launaea fragilis*.

**Figure 7.** The main highly palatable species (palatability index (PI) = 5) recorded in the arid and desert rangelands of Tataouine, Tunisia (spring 2018) (Photos by Mouldi Gamoun).

Additionally, rangelands are characterized by a large number of palatable species belonging to Poaceae, such as *Oloptum miliaceum*, *Dactylis glomerata*, *Stipagrostis plumosa*, *Cynodon dactylon*, and *Cenchrus ciliaris*. Annual plants that grow mainly in spring are an important component of arid rangelands. They include, for example, many species that are very palatable, such as *Astragalus asterias*, *Astragalus hamosus*, *Astragalus arpilobus*, *Sulla carnosa*, *Hippocrepis areolata*, and *Medicago laciniata* of the Fabaceae.

Moreover, in addition to its pastoral value, these rangelands have many valuable medicinal and aromatic plants [61]. Many traditional herbal remedies derived from pastoral plants that originated in natural rangelands have become modern medicines. [62] described the medicinal value of more than 45 species spread over 22 botanical families native to arid and desert rangelands of Tunisia. These species include the following (Figure 8): *Allium roseum, Allium ampeloprasum, Searsia tripartita, Periploca angustifolia, Capparis spinosa, Herniaria fontanesii, Haloxylon scoparium, Artemisia herba-alba, Artemisia campestris, Henophyton deserti, Diplotaxis harra, Citrullus colocynthis, Ephedra alata, Ephedra altissima, Hyparrhenia hirta, Rosmarinus officinalis, Thymus algeriensis, Thymbra capitata, Ajuga iva, Daucus carota, Marrubium deserti, Teucrium polium, Calicotome villosa, Retama raetam, Cymbopogon schoenanthus, Calligonum polygonoides, Polygonum equisetiforme, Ziziphus lotus, Thymelaea hirsuta, Thapsia garganica, Deverra denudata, Deverra tortuosa, Nitraria retusa, Peganum harmala*, and *Zygophyllum album*.

Some species have high culinary value [62], such as *Rosmarinus officinalis, Thymus algeriensis, Capparis spinosa, Allium roseum,* and *Allium ampeloprasum*. Some species, such as *Ziziphus lotus* and *Nitraria retusa*, have small edible fruits used by the local population [63]. A few species are poisonous or toxic to animals, such as *Peganum harmala, Euphorbia terracina, Euphorbia retusa*, and *Adonis microcarpa*.

The floristic survey conducted on the rangelands revealed a high number of endangered species. The greatest threat to most species is overgrazing, plant eradication, and degradation of their habitat. The endangered forage species were *Anabasis oropediorum, Anarrhinum fruticosum, Calligonum polygonoides, Echiochilon fruticosum, Eragrostis papposa, Sulla carnosa, Stipa parviflora,* and *Stipa lagascae*. Likewise, the majority of the medicinal plants are considered seriously threatened due to overuse. For example, *Allium roseum* is threatened with extinction because of overexploitation; the harvesting of this plant is very destructive because the bulbs are torn off during harvesting [64]. Of the 35 medicinal species recorded in our survey, eight are classified as critically endangered: *Allium roseum, Allium ampeloprasum, Ephedra alata, Ephedra altissima, Rosmarinus officinalis, Thymus algeriensis, Thymbra capitata*, and *Cymbopogon schoenathus*.

Out of the 279 species that have been identified, five species are endemic to the country, which are *Anarrhinum fruticosum* subsp. *brevifolium, Helianthemum virgatum* subsp. *africanum, Limonium tunetanum, Onopordum espinae*, and *Teucrium Alopecurus* [65]. This inventory has devoted a great deal of attention to the botanical composition and species diversity of arid and desert rangelands in Tataouine that provide important ecosystem services, yet they are still neglected.

In fact, these rangelands have been subjected to intensive anthropogenic and climatic disturbances, such as overgrazing and drought over a long period of time, and their overall condition is deteriorating [1–3]. Nevertheless, a great number of plant species are still vitally important for human health, as well as livestock and wildlife feeding. The recorded wide range of species reflects the significant resilience and adaptation of these arid rangeland ecosystems. Among the strategies adopted by the arid plants to overcome such harsh conditions is their ability to go dormant and cope with extreme heat and recurrent drought to ensure that neither internal temperatures nor tissue dehydration reach low levels [66].

The degree of floristic importance varies from one species to another and is based on spatial distribution across the region. Interesting enough, south Tunisia's flora includes 5 species endemics to the country that are classified as endangered. The important botanical composition recorded during this study of 279 species, compared to 135 species that were identified during an average rainy year and only 60 species during a dry year [9,16], is regarded as a significant indicator of rangeland resilience and their ability to recover from drought.

There is solid evidence that greater botanical diversity is essential to sustainable land use by increasing forage yield, pollinators, as well as weed and pest suppression [67]. Certainly, high botanical diversity also plays a key role in soil aggregate stability. The root

system of the plant improves soil structure and increases the soil organic matter [68]. Furthermore, plant diversity and root traits also benefit essential soil physical properties [69]. In recent years, advanced research in ecological conservation, combined with greater focus on ecosystem services, have enhanced our understanding of these complex ecosystems but also highlighted several challenges, calling for innovative measures to preserve our natural resources.

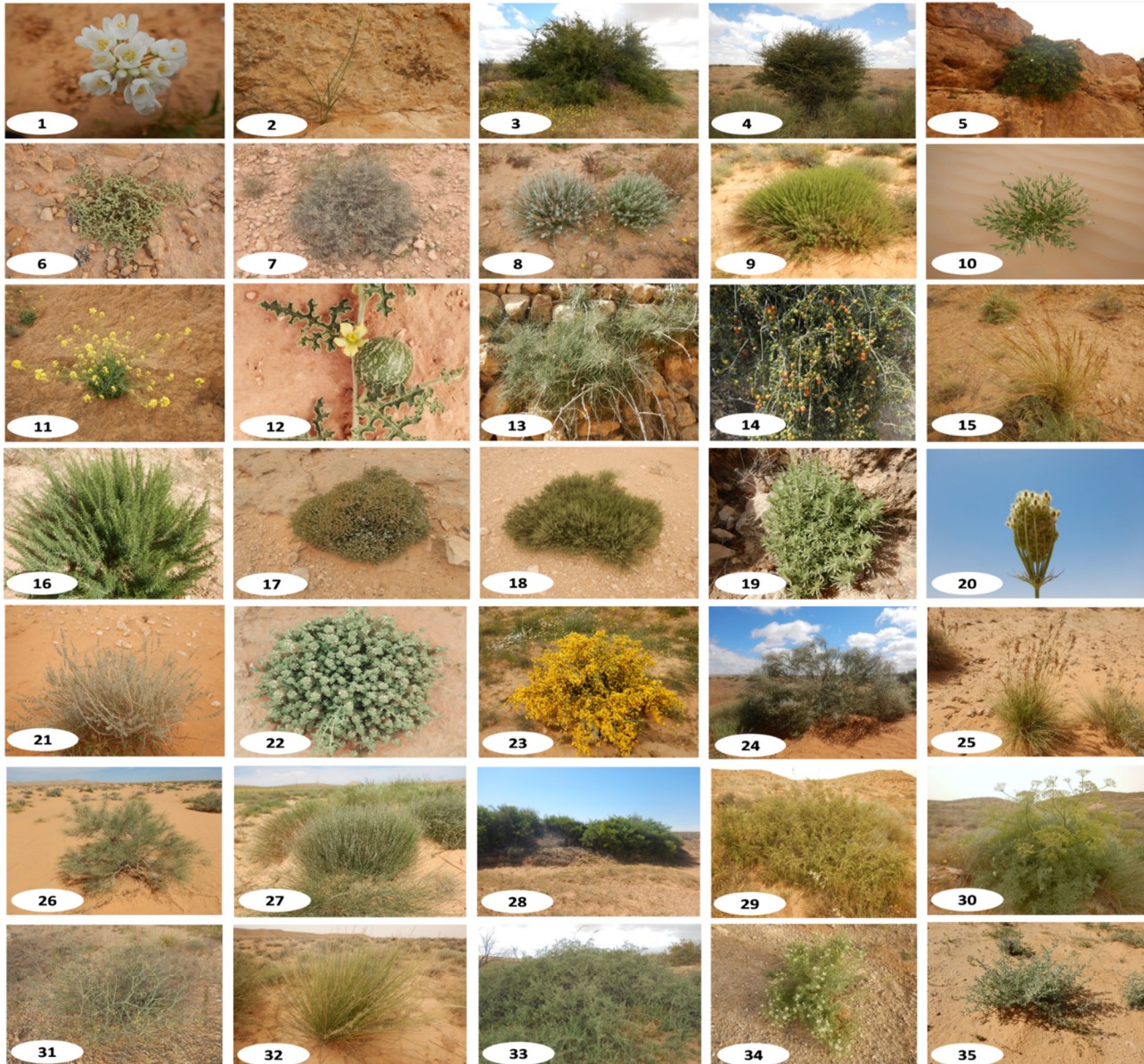

1) *Allium roseum*, 2) *Allium ampeloprasum*, 3) *Searsia tripartita*, 4) *Periploca angustifolia*, 5) *Capparis spinosa*, 6) *Herniaria fontanesii*, 7) *Hammada scoparia*, 8) *Artemisia herba-alba*, 9) *Artemisia campestris*, 10) *Henophyton deserti*, 11) *Diplotaxis harra*, 12) *Citrullus colocynthis*, 13) *Ephedra alata*, 14) *Ephedra altissima*, 15) *Hyparrhenia hirta*, 16) *Rosmarinus officinalis*, 17) *Thymus algeriensis*, 18) *Thymbra capitata*, 19) *Ajuga iva*, 20) *Daucus carota*, 21) *Marrubium deserti*, 22) *Teucrium polium*, 23) *Calicotome villosa*, 24) *Retama raetam*, 25) *Cymbopogon schoenanthus*, 26) *Calligonum polygonoides*, 27) *Polygonum equisetiforme*, 28) *Ziziphus lotus*, 29) *Thymelaea hirsuta*, 30) *Thapsia garganica*, 31) *Deverra tortuosa*, 32) *Deverra denudata*, 33) *Nitraria retusa*, 34) *Peganum harmala*, 35) *Zygophyllum album*.

**Figure 8.** Medicinal and aromatic plants recorded in the arid and desert rangelands of Tataouine, Tunisia (spring 2018) (Photos by Mouldi Gamoun).

## 4. Conclusions

In addition to the ecological importance of safeguarding the stability of the natural environment, the rich and diverse flora of arid and desert rangelands in Tunisia provide essential ecological services to the livestock and human population. They provide a great

variety of native forage and medicinal plants with modern pharmacological uses. Because the rangelands are not protected, a serious threat to floral diversity, caused by human activity, has occurred in a large area of the arid rangelands, while climate conditions are creating significant transformation through favorable years. Although the rangelands of Tataouine are dry, they are the native habitat of more than 10% of the total flora of Tunisia. The recorded species are mainly annuals and perennials characteristic of dry ecosystems. The main families are the Asteraceae, Poaceae, Fabaceae, Amaranthaceae, Brassicaceae, Boraginaceae, Caryophyllaceae, Lamiaceae, Apiaceae, and Cistaceae, which together account for 65% of the flora. Therophytes comprised the highest number of species, followed by chamaephytes and hemicryptophytes. These rangelands are rich in foraging species of high nutritional value for livestock feeding and many important plants used in both traditional and modern medicine. Despite this significant floristic richness, certain species remain endangered and must be effectively managed and protected to avoid their extinction. For this, it is necessary to set up a comprehensive biodiversity conservation program. This program should include sustainable rangeland management that expands the protected areas containing suitable habitats of rare and endemic species. Furthermore, it would be wise to establish botanical gardens or field gene banks as part of a long-term biodiversity conservation program for endangered species.

**Supplementary Materials:** The following are available online at https://www.mdpi.com/2073-445X/10/3/313/s1.

**Author Contributions:** Conceptualization, M.G.; methodology, M.G. and M.L.: validation, M.L.: formal analysis, M.G.: investigation, M.G. and M.L.; data curation, M.G.; writing—original draft preparation, M.G.; writing—review and editing, M.G. and M.L.; visualization, M.G. and M.L.; supervision, M.L.; project administration, M.L.; funding acquisition, M.L. Both the authors have read and agreed to the published version of the manuscript.

**Funding:** This research was funded by the CGIAR Research Program on Livestock—Livestock and Environment Flagship, agreement number 200173, and the APC was also funded by the same source of funding.

**Institutional Review Board Statement:** Not applicable.

**Informed Consent Statement:** Not applicable.

**Data Availability Statement:** All data are available upon request.

**Acknowledgments:** This work was supported by the International Center for Agricultural Research in the Dry Areas (ICARDA), the Office of Livestock and Pastures (OEP, Tunisia), and the CGIAR Research Program on Livestock (CRP Livestock), led by the International Livestock Research Institute (ILRI).

**Conflicts of Interest:** The authors declare no conflict of interest

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
