# Peer review of "Botanical Composition and Species Diversity of Arid and Desert Rangelands in Tataouine, Tunisia"

_land, doi:10.3390/land10030313_

Round 1

Reviewer 1 Report

The manuscript entitled “Botanical composition and species diversity of arid and desert rangelands in Tataouine, Tunisia” give information about the botanical composition, diversity, life form habitat class, palatability of plants growing in arid rangelands from southern Tunisia, region Tataouine. The authors states that this governorate covering an area of 38,889 km2, consists in 38 % from rangelands (about 1500000 ha).

The study aim consists in a better understanding and “document knowledge concerning species richness and botanical composition”, and after that the authors try to link these indicators with an exceptional rainy season in the region. The objectives of the study are too vague. I strongly recommend to better formulate the objectives of the study.  Furthermore, the linkage with the rainy season cannot be understood by the readers because the manuscript presents only the botanical composition of rangelands after 2017/2018 season, which was rainy, but not also after a dryer season, to can make a comparison. Therefore, from the present study no documented conclusion can be drawn about the impact of water availability on species richness s the authors suggest in L82-85.

Regarding the methodology it is not specified how the species inventorying is due. I wonder that the inventory was carry out on the whole rangeland of Tataouine (1500000 ha). Therefore, more information about the inventorying is needed.

Fig. 2 shows the annual and monthly variability during the season 2017/2018, how the annual value from a single season can vary in each month?

L116-117: What means: the growth of herbaceous plants… was spectacular?

L129: greater than what?

I wonder that by the Land journal can be admitted having the Results together with the Discussion.

A conclusion at final of the Discussion is missing.

In the present for, the Manuscript does not meet the scientific and formatting requirements of Land journal. Without a clear ecological research question, the topic of the Manuscript seems to be more appropriate to a journal focusing of botanical studies.

Author Response

Comment 1

The manuscript entitled “Botanical composition and species diversity of arid and desert rangelands in Tataouine, Tunisia” give information about the botanical composition, diversity, life form habitat class, palatability of plants growing in arid rangelands from southern Tunisia, region Tataouine. The authors states that this governorate covering an area of 38,889 km2, consists in 38 % from rangelands (about 1500000 ha).

The study aim consists in a better understanding and “document knowledge concerning species richness and botanical composition”, and after that the authors try to link these indicators with an exceptional rainy season in the region. The objectives of the study are too vague. I strongly recommend to better formulate the objectives of the study.  Furthermore, the linkage with the rainy season cannot be understood by the readers because the manuscript presents only the botanical composition of rangelands after 2017/2018 season, which was rainy, but not also after a dryer season, to can make a comparison. Therefore, from the present study no documented conclusion can be drawn about the impact of water availability on species richness s the authors suggest in L82-85.

Regarding the methodology it is not specified how the species inventorying is due. I wonder that the inventory was carry out on the whole rangeland of Tataouine (1500000 ha). Therefore, more information about the inventorying is needed.

Response

The objective of the study has been added.

For the methodology we covered almost the entire rangeland areas. As it is impossible for us to know when and where these rare species would appear, we relied on expert knowledge (including herders who practice transhumance as well as elderly pastoralists) to guide us to the exact locations where certain species are resurfacing during such an exceptional favorable year.  

Comment 2

Fig. 2 shows the annual and monthly variability during the season 2017/2018, how the annual value from a single season can vary in each month?

Response:

Figure 2 is corrected showing the monthly rainfall variability

Comment 3

L116-117: What means: the growth of herbaceous plants… was spectacular?

Response

It is corrected

Comment 4

L129: greater than what?

Response

It is corrected

Comment 5

L129: greater than what?

Response

It is corrected

Comment 6

A conclusion at final of the Discussion is missing.

Response

A conclusion is added

Reviewer 2 Report

The work deals with the grasslands and scrublands of arid places in southern Tunisia, and their use as medicinal, culinary and livestock plants. This work can and should be improved. The study area is interesting in that it presents habitats of special interest, therefore it is necessary for the authors to place greater emphasis on the state of conservation of the habitats.

1.-It is suggested to add a table to a column in which it is stated whether the plant is endemic or not, and another column with the habitat in which the species is found, indicating those of special protection.

2.- The title of table 1 does not say anything, it must be modified.

3.- All species must bear their authorship the first time they are cited, later it is not necessary.

4.- They must control the synonymy of the species, the flora used from North Africa is very old, but I also do not see it in references, we suggest using the floras of Valdes et al. for Morocco, Flora Ibérica.

5.- The feet of the figures are too small and do not look good.

6.- Please check the writing of the subspecies, they are written in lowercase, the figure says Helianthemum virgatum subsp. Afrocanum, must put africanum

7.-In line 149 it says Oryzopsis mialiacea, Piptatherum miliaceum, check it because they are synonyms.

8.- In the section of plants used in medicine and as culinary, you must put references.

Author Response

Comment 1

It is suggested to add a table to a column in which it is stated whether the plant is endemic or not, and another column with the habitat in which the species is found, indicating those of special protection.

Response

One column for endemism was added as requested. Yet we believe since they are only 5 endemic species compared to over 270 species

Comment 2

The title of table 1 does not say anything, it must be modified.

Response

The tile of table 1 was modified

Comment 3

The title of table 1 does not say anything, it must be modified.

Response

The tile of table 1 was modified

Comment 4

All species must bear their authorship the first time they are cited, later it is not necessary.

Response

The authorship of species are cited only in table 1

Comment 5

They must control the synonymy of the species, the flora used from North Africa is very old, but I also do not see it in references, we suggest using the floras of Valdes et al. for Morocco, Flora Ibérica.

Response

Synonymy of the species are updating using the synonymic index of the flora of North Africa of Dobignard &  Chatelain 2010–2013 and Plants of the World Online (POWO).

Comment 6

The feet of the figures are too small and do not look good.

Response

Figures have been modified

Comment 7

Please check the writing of the subspecies, they are written in lowercase, the figure says Helianthemum virgatum subsp. Afrocanum, must put africanum

Response

Corrected

Comment 8

In line 149 it says Oryzopsis mialiacea, Piptatherum miliaceum, check it because they are synonyms.

Response

Corrected

Comment 8

In the section of plants used in medicine and as culinary, you must put references.

Response

References were added

Reviewer 3 Report

First of all, it is striking that the authors had sent the manuscript of an article, which is clearly included in the botanical discipline, to a journal such as Land, whose specialty is: land use/land change, land management, land system science, landscape, soil-sediment-water systems, urban contexts and urban-rural interactions, and land–climate interactions, etc. I cannot see what is the justification for including this work in Land.

In the second place, it is not clear what the objective of the evaluated work is. The authors point out (lines 45-47): “We conducted this study in an attempt to better understand and document knowledge concerning species richness and botanical composition”, but in their work they do not indicate localities, nor dates of collection, nor conservation status of the populations (among other botanical relevant information). So I do not know how they will contribute to knowledge concerning to species richness and botanical composition.

Third, there is an absolute lack of methodology in the development of the work, which completely invalidates the information provided. There are barely 18 lines of text in the Materials and Methods section. Of which 11 lines correspond to the subsection Study Area. In this subsection geographical, climatic (line 56-57), geological and edaphic (lines 57-58) and land use data are provided in a more than succinct manner. The other subsection, Data Collection, is even more disappointing, as it is limited to only 7 lines. Here the authors do not indicate how the names of the taxa that appear in the evaluated work have been obtained. What do these names refer to, herbarium sheets, bibliographic citations, location in GBIF...? Neither the climatic variables are detailed, nor the measures of palatability, nor how the medicinal plants have been categorized, nor are aspects related to conservation developed. Only, the authors mention a work used for them as a reference for the nomenclature of the species present in the manuscript (lines 68-69). However, that reference corresponds to a technical document, it is not a flora, which would be the pattern for a botanical work.

Fourth, there is no separation between results and discussion. In this mix-section the authors do not provide results, they directly develop discussions on various issues. The problem is that in a rigorous scientific text, the discussion has to be based on the results obtained. Thus, the authors discuss the role of the rainfall regime on the dynamics of the vegetation (lines 77-85), without providing any data in this regard; on the dominance of certain biotypes in degraded places, without providing any data in this regard either (lines 96-98); on germination (line 98; the importance of habitats and microhabitats (lines 99-110); the palatability of the species (124-129); etc. All unsupported by information provided by the results of the work.

Fifth, figures 7 and 8 are superfluous. They contribute nothing. Table 1 is very extensive and should not appear in the text, it should appear in an appendix.

Sixth, the conclusions based on the premises developed in this manuscript are erroneous, as they lack rigor and scientific methodology.

Taking into account all of the above, my recommendation is the evaluated manuscript should be rejected for publication in Land

Author Response

Comment 1

First of all, it is striking that the authors had sent the manuscript of an article, which is clearly included in the botanical discipline, to a journal such as Land, whose specialty is: land use/land change, land management, land system science, landscape, soil-sediment-water systems, urban contexts and urban-rural interactions, and land–climate interactions, etc. I cannot see what is the justification for including this work in Land.

Response 1

Originally this paper was meant to be published in the Special Issue "Drylands and Deserts: Research Intersections" led by the guest editor Dr. Troy Sternberg.

We sent the abstract to Dr. Troy Sternberg and we got a positive reply to go ahead and submit the manuscript. Later, we were informed by the journal that the Guest editor of the Special Issue that paper belonged to quit, and the journal decided to move the paper as regular.

Comment 2

In the second place, it is not clear what the objective of the evaluated work is. The authors point out (lines 45-47): “We conducted this study in an attempt to better understand and document knowledge concerning species richness and botanical composition”, but in their work they do not indicate localities, nor dates of collection, nor conservation status of the populations (among other botanical relevant information). So, I do not know how they will contribute to knowledge concerning to species richness and botanical composition.

Response 2

This has been addressed in the objective and methodology.

Comment 3

Third, there is an absolute lack of methodology in the development of the work, which completely invalidates the information provided. There are barely 18 lines of text in the Materials and Methods section. Of which 11 lines correspond to the subsection Study Area. In this subsection geographical, climatic (line 56-57), geological and edaphic (lines 57-58) and land use data are provided in a more than succinct manner. The other subsection, Data Collection, is even more disappointing, as it is limited to only 7 lines. Here the authors do not indicate how the names of the taxa that appear in the evaluated work have been obtained. What do these names refer to, herbarium sheets, bibliographic citations, location in GBIF...? Neither the climatic variables are detailed, nor the measures of palatability, nor how the medicinal plants have been categorized, nor are aspects related to conservation developed. Only, the authors mention a work used for them as a reference for the nomenclature of the species present in the manuscript (lines 68-69). However, that reference corresponds to a technical document, it is not a flora, which would be the pattern for a botanical work.

Response 3

Methodology has been consolidated by adding more details how, when and where the inventory was conducted.

Comment 4

Fourth, there is no separation between results and discussion. In this mix-section the authors do not provide results, they directly develop discussions on various issues. The problem is that in a rigorous scientific text, the discussion has to be based on the results obtained. Thus, the authors discuss the role of the rainfall regime on the dynamics of the vegetation (lines 77-85), without providing any data in this regard; on the dominance of certain biotypes in degraded places, without providing any data in this regard either (lines 96-98); on germination (line 98; the importance of habitats and microhabitats (lines 99-110); the palatability of the species (124-129); etc. All unsupported by information provided by the results of the work.

Response 4

More clarification was added.

Comment 5

Fifth, figures 7 and 8 are superfluous. They contribute nothing. Table 1 is very extensive and should not appear in the text, it should appear in an appendix.

Response 5

Figures 7 and 8 could be easily deleted if decided not worth including. However, these figures could be of great interest to readers to visualize how each species look like and help them in identifying certain rare species.

We fully agree that table 1 is too long and we have included it as an appendix.

Comment 6

Sixth, the conclusions based on the premises developed in this manuscript are erroneous, as they lack rigor and scientific methodology.

Response 6

Conclusion has been strengthened

Round 2

Reviewer 1 Report

The authors improved significantly their manuscript entitled “Botanical composition and species diversity of arid and desert rangelands in Tataouine, Tunisia”, considering my recommendations.

Therefore, I recommended to accept it to publication.

I have only some minor comments:

L55: perhaps could delete “eventually”.

L99-100: check if the sentence is correct.

L120: different not dif-ferent

L122 on instead one

Author Response

Comment 1

L55: perhaps could delete “eventually”.

Response: The word “eventually” is deleted

Comment 2

L99-100: check if the sentence is correct.

Response: the sentences is corrected

Comment 3

L120: different not dif-ferent

Response: The word is corrected

Comment 4

L122 on instead one

Response: The word is corrected

Reviewer 2 Report

The subspecie africanum is still capitalized, I imagine it will be a typographical error, in the printing tests please correct it.

Author Response

Comment 1

The subspecie africanum is still capitalized, I imagine it will be a typographical error, in the printing tests please correct it

Response: Thank you very much, the name is corrected and it is not capitalized anymore

Reviewer 3 Report

I still find it very strange that a work whose objective is a floristic analysis of a territory is sent to a journal such as Land, whose specialty is: land use/land change, land management, land system science, landscape, soil-sediment-water systems, urban contexts and urban-rural interactions, and land–climate interactions, etc.

The evaluated manuscript, in addition to lacking a rigorous methodology, does not provide information of interest and, furthermore, its publication would introduce an enormous amount of noise, by disclosing data that are not verifiable.

In the previous evaluation, being concerned about the major problems that I noticed in the manuscript, I did not comment on some inadequate details of the evaluated text. Such errors, in comparison with the aforementioned problems, were minor, but now in this second evaluation I want to highlight them.

For example, in the Introduction section, there are no references to previous works on flora, chorology, or taxonomy, which provide floristic data on the study area. In such a way that, it seems that the authors of the manuscript are the first to provide floristic information about these places. However, this is not the case. Numerous references are available that provide data on the presence of plant species in the studied territory. On the other hand, the authors comment on some ideas about biodiversity, the negative effects of grazing or drought, or the goods and services provided by plants, which, although may be interesting, they do not are consistent whit the data provided in this manuscript (shown in Annex 1).

It should also be noted that in this second version the authors have defined with more precision the objective of the work: “Specifically, the objectives of our study are to examine the botanical composition including plant family, life form, habitat class, palatability, and medicinal and aromatic plants and to determine the relevance of arid and desert rangeland” (lines 51-52).

Regarding the Material & Methods section, the main problem of this manuscript continues to be the lack of methodology and scientific rigor, reflected in the poor and inoperative development of this section. The data related to climate, geomorphology, edaphology are still very scarce. Besides, there is no indication of how palatability has been assessed, nor on what basis different species of the flora have been classified as threatened, nor how medicinal properties have been attributed to some of the species, among other facts.

A capital and crucial methodological aspect is that researches who work with plants (as well as scientists from other disciplines) must ensure that the data they provide are truly verifiable. To this end, specimens must be deposited in herbaria so that the identity of the samples can be verified. In this way, data related to these species will meet scientific standards. In relation to this, the authors wrote (lines 83-85 state): "All species were photographed using a high-resolution digital camera showing structure, leaf, stem, flower and fruit if existing". This introduces a serious problem of uncertainty, since species identification, for many of the vascular plant genera listed in Annex 1, requires a high level of taxonomic expertise and it is impossible to make a correct determination from a photograph. Likewise, where are these photos and who has access to them? Therefore, in this manuscript, knowing with certainty the identity of the plants listed in Annex 1 becomes an act of faith, because it is impossible to scientifically verify their identity.

Regarding the Results and Discussion section, although the authors have deleted several statements that were not correct, the section remains confusing. So that it is difficult to distinguish between what may be results contributed by the authors and what are comments regarding bibliographic references. Similarly, this section continues to include some discussions about information that has not been obtained in the way the methods are described in the Material & Methods section. For example, comments on truffles, comments on the pastoral value of some plants, their culinary value, medicinal value, endemicity, threatened species, or the effects of overgrazing, drought or climate on the studied plants.

Figures 7 and 8 are totally superfluous, they contribute nothing. In addition, the quality of the images is too low to appreciate any detail. Por tanto, el trabajo en su actual.

As in the previous version of the manuscript, the conclusions based on the premises developed in this manuscript are erroneous, as they lack rigor and scientific methodology. Furthermore, only the conclusions related to the floristic analysis respond to the logic of a scientific work, with the objectives stated in the introduction and developed (poorly) in the Material & Methods section, the rest are pointless conclusions.

Having evaluated this new version of the manuscript, and taking into account all of the above, my recommendation is to reject the manuscript for publication in Land

Author Response

Reviewer 3 response

I still find it very strange that a work whose objective is a floristic analysis of a territory is sent to a journal such as Land, whose specialty is: land use/land change, land management, land system science, landscape, soil-sediment-water systems, urban contexts and urban-rural interactions, and land–climate interactions, etc.

Response: We have responded to the reviewer comments at the first round

The evaluated manuscript, in addition to lacking a rigorous methodology, does not provide information of interest and, furthermore, its publication would introduce an enormous amount of noise, by disclosing data that are not verifiable.

In the previous evaluation, being concerned about the major problems that I noticed in the manuscript, I did not comment on some inadequate details of the evaluated text. Such errors, in comparison with the aforementioned problems, were minor, but now in this second evaluation I want to highlight them.

For example, in the Introduction section, there are no references to previous works on flora, chorology, or taxonomy, which provide floristic data on the study area. In such a way that, it seems that the authors of the manuscript are the first to provide floristic information about these places. However, this is not the case. Numerous references are available that provide data on the presence of plant species in the studied territory. On the other hand, the authors comment on some ideas about biodiversity, the negative effects of grazing or drought, or the goods and services provided by plants, which, although may be interesting, they do not are consistent whit the data provided in this manuscript (shown in Annex 1).

Response: We have added more information in the introduction  

It should also be noted that in this second version the authors have defined with more precision the objective of the work: “Specifically, the objectives of our study are to examine the botanical composition including plant family, life form, habitat class, palatability, and medicinal and aromatic plants and to determine the relevance of arid and desert rangeland” (lines 51-52).

Regarding the Material & Methods section, the main problem of this manuscript continues to be the lack of methodology and scientific rigor, reflected in the poor and inoperative development of this section. The data related to climate, geomorphology, edaphology are still very scarce. Besides, there is no indication of how palatability has been assessed, nor on what basis different species of the flora have been classified as threatened, nor how medicinal properties have been attributed to some of the species, among other facts.

Response: We have responded to the reviewer comments at the first round

A capital and crucial methodological aspect is that researches who work with plants (as well as scientists from other disciplines) must ensure that the data they provide are truly verifiable. To this end, specimens must be deposited in herbaria so that the identity of the samples can be verified. In this way, data related to these species will meet scientific standards. In relation to this, the authors wrote (lines 83-85 state): "All species were photographed using a high-resolution digital camera showing structure, leaf, stem, flower and fruit if existing". This introduces a serious problem of uncertainty, since species identification, for many of the vascular plant genera listed in Annex 1, requires a high level of taxonomic expertise and it is impossible to make a correct determination from a photograph. Likewise, where are these photos and who has access to them? Therefore, in this manuscript, knowing with certainty the identity of the plants listed in Annex 1 becomes an act of faith, because it is impossible to scientifically verify their identity.

Response: this study is a part of Tunisian Digital herbarium that we are in the process to develop which is still under process therefore we assure that we have documented all the studies species with consultancy of many taxonomist specialists

Regarding the Results and Discussion section, although the authors have deleted several statements that were not correct, the section remains confusing. So that it is difficult to distinguish between what may be results contributed by the authors and what are comments regarding bibliographic references. Similarly, this section continues to include some discussions about information that has not been obtained in the way the methods are described in the Material & Methods section. For example, comments on truffles, comments on the pastoral value of some plants, their culinary value, medicinal value, endemicity, threatened species, or the effects of overgrazing, drought or climate on the studied plants.

Response: thank you for the comments we have addreesed most of these nots at the first round

Figures 7 and 8 are totally superfluous, they contribute nothing. In addition, the quality of the images is too low to appreciate any detail. Por tanto, el trabajo en su actual.

Response: The Figures are replaced by better resolution ones.

As in the previous version of the manuscript, the conclusions based on the premises developed in this manuscript are erroneous, as they lack rigor and scientific methodology. Furthermore, only the conclusions related to the floristic analysis respond to the logic of a scientific work, with the objectives stated in the introduction and developed (poorly) in the Material & Methods section, the rest are pointless conclusions.

Response: Thank you.  

Having evaluated this new version of the manuscript, and taking into account all of the above, my recommendation is to reject the manuscript for publication in Land
